# Stress Response Behavior, Microstructure Evolution and Constitutive Modeling of 22MnB5 Boron Steel under Isothermal Tensile Load

**Qian Zhou [1], Pengcheng Guo [2],\* and Feng Qin [1]**

[1] College of Materials Science and Engineering, Hebei University of Engineering, Handan 056038, China; dqianzhou@163.com (Q.Z.); drfqin@163.com (F.Q.)

[2] College of Mechanical and Electrical Engineering, Central South University of Forestry and Technology, Changsha 410004, China

\* Correspondence: gpch860429@163.com

**Abstract:** 22MnB5 boron steel has become one of the main choices for lightweight vehicles due to its extremely high mechanical properties. To explore the intrinsic relationship between the thermoforming process and thermo-mechanical behavior for constitutive modeling and thermoforming of vehicle structure, thermal tensile tests in wide ranges of deformation temperature (500 °C to 950 °C) and strain rate (0.01 s$^{-1}$ to 10 s$^{-1}$) were performed using a Gleeble-1500D thermal simulator with hot-rolled 22MnB5 boron steel. With increasing applied strain and strain rate, the flow stress increases gradually and then tends to saturation after reaching peak stress, except for that at 0.01 s$^{-1}$ and 500 °C. With increasing deformation temperature, the microstructure transforms from a mixture of bainite, ferrite and pearlite to lath-shaped martensite accompanied with some residual austenite. At 950 °C, the average size of martensite decreases with increasing applied strain rate. After thermoforming with austenitizing temperature of 950 °C, lath-shaped martensite accompanied with some residual austenite is obtained in a thermoformed U-shaped structural part, resulting in a dramatical increase in tensile strength. In contrast, the tensile strength of sidewall is slightly higher than that of bottom. Based on the Arrhenius-type constitutive model, a modified constitutive model is constructed with a relative error of less than 5%, which can well describe the flow stress behavior of the studied 22MnB5 boron steel.

**Keywords:** 22MnB5 boron steel; stress response behavior; microstructure evolution; constitutive model; thermoforming

## 1. Introduction

In recent years, environmental problems caused by the rapid increase in the number of vehicles have become more and more prominent; therefore, relevant laws have been promulgated by governments around the world. It is well known that lightweight is an important way to reduce the fuel consumption of vehicles [1,2]. Due to its extremely high mechanical properties, boron steel has become one of the main choices for constructing lightweight vehicles [3–5]. However, parts made by high-strength steels are difficult to form by cold stamping, and the resulting spring back is severe [6]. Stepwise thermoforming can effectively solve the above problem, and is considered to be the most suitable method for obtaining high-strength structural parts with complex shapes [7,8].

At present, research on boron steels mainly focuses on the relationship among deformation parameters, stress response behavior, and microstructure evolution [9,10]. The flow stress and the slope of strain hardening decrease with increasing deformation temperature, while they increase with increasing strain rate [11]. The decrease in cooling rate brings about an increase in $M_s$ and $M_f$ temperatures when another phase has been formed before martensite transformation starts, while it causes a reduction if the final microstructure

contain smartensite only [9]. In contrast, the reduction of cooling rate only leads to an increase in $B_s$ in the undeformed condition, while the $B_s$ is a constant in the hot deformed condition [9]. As reported, the evolved austenite strongly affects the phase transition energy and mechanical properties [9,12]. As deformation temperature increases from 25 °C to 450 °C, the volume fraction of martensite in quenched samples decreases from 83% to 40%, while it increases from 16% to 52% for bainite, resulting in a decrease in tensile strength from 1454 MPa to 963 MPa and an increase in total elongation from 6.6% to 7.9% [13]. A fine lath-shaped microstructure consisting of martensite and/or bainite with undissolved carbides is acquired by rapid heating followed by isothermal quenching, which has been used as one of the strengthening methods for 22MnB5 boron steels [14]. It has been reported that, when stretched at high temperature, a bell-shaped strain distribution can be detected before maximum tensile load, and the stress response behavior and fracture strain are influenced by the gauge length of the samples, resulting in a deviation of more than 16% [15].

Based on an Arrhenius equation, the parameters of the 22MnB5 steel are calculated on the basis of the data obtained by thermal stretch at temperatures of 700, 800, and 900 °C and strain rates of 0.01, 0.1, and 1 s$^{-1}$ [10].To describe the flow behavior more accurately, a strain-compensated Arrhenius-type constitutive model and a set of unified viscoplastic equations have been established for B1500HS boron steel that take into account the effect of dislocation density [16,17]. In comparison, the former exhibits a slightly higher prediction accuracy over a wide temperature range [17]. To describe the dependency of saturation stress and yield stress on strain rate and temperature, an optimized model was derived by absorbing the Kocks model into the Voce formulation [18]. In addition, a modified Fiel-Backofen model containing strain, strain rate and volume fraction of quenched microconstituent has been established that can accurately predict the stress response behavior of quenched boron steels [13,16]. Furthermore, based on thermal compression tests at deformation temperatures of 800~950 °C and strain rates of 0.01~0.8 s$^{-1}$, a physical constitutive model considering strain hardening, dynamic recovery (DRV) and dynamic recrystallization (DRX) is developed with a correlation coefficient of 0.997 and an average absolute relative error of 3.89% [19]. Phenomenological constitutive models are different from mechanism-based constitutive equations, which are usually experiential and intuitive, and do not require an in-depth understanding of the physical phenomena. However, to date, most models for boron steel have focused on hardening behavior, while the softening effect caused by DRV and DRX is neglected.

On the whole, there have been some studies on the mechanical behavior and constitutive modeling of 22MnB5 boron steel, while the ranges of strain rate and deformation temperature involved in previous studies are relatively narrow, and there are almost no related studies in corporating the thermoforming of vehicle body structures. In addition, as mentioned previously, the thermoforming process determines the microstructure and mechanical properties of structural parts [20]. Moreover, the mechanical constitutive equation is one of the most necessary mathematical models for the thermoforming simulation of boron steel. Therefore, it is extremely necessary to study the flow stress behavior of the 22MnB5 steel in wide ranges of deformation temperature and strain rate in order to explore the intrinsic relationship between thermoforming process and thermo-mechanical behavior for constitutive modeling and optimization of forming process, and to carry out thermoforming tests related to vehicle body. In this paper, thermal tensile tests were conducted in wide ranges of deformation temperature and strain rate using a Gleeble 1500D thermo-mechanical simulator with hot-rolled 22MnB5 boron steel, and the stress response behavior and microstructure evolution under various conditions were studied. On this basis, thermoforming tests were carried out with a structural part related to vehicle body.

## 2. Materials and Methods

Hot-rolled 22MnB5 boron steel with thickness of 2.0 mm was used in the present study, which was coated with hot-dip Al-Si before thermal tensile and thermoforming tests. The

chemical composition (wt. %) is listed in Table 1. Thermal tensile samples (Figure 1a) with gauge length of 30 mm and width of 10 mm were cut from the as-received sheets along the rolling direction.

**Table 1.** Chemical component of the 22MnB5 boron steel (wt. %).

| C | Mn | Si | Al | Ti | Cr | B | Fe |
|------|------|------|------|------|------|--------|------|
| 0.22 | 1.20 | 0.20 | 0.02 | 0.03 | 0.20 | 0.0035 | Bal. |

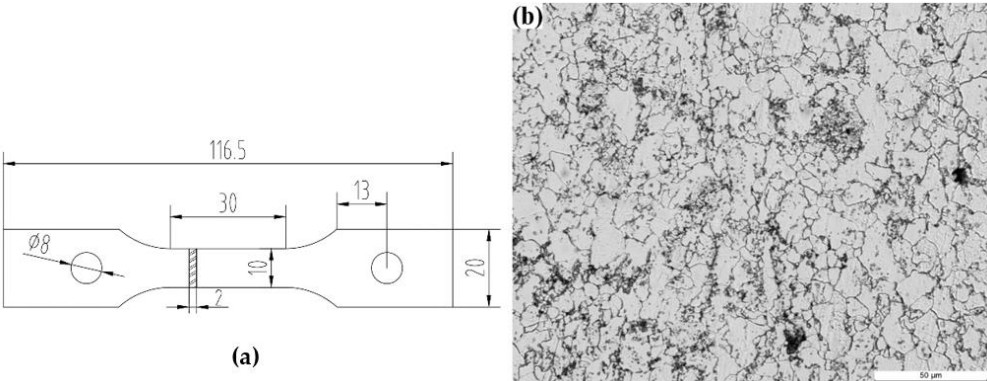

**Figure 1.** (**a**) Dimensions of thermal tensile specimen (mm) and (**b**) initial OM image of the studied 22MnB5 boron steel.

Thermal tensile tests were carried out using a Gleeble-1500D thermal simulator (DSI, St. Paul, MN, USA) at 500, 650, 700, 800 and 950 °C with strain rates of 0.01, 0.1, 1 and $10\,s^{-1}$. The heating and cooling of samples were realized by heat conduction at the clamping end. First, specimen was heated to 950°C with a rate of 15 °C/s followed by soaking with 180 s for complete austenitizing. Then, it was rapidly cooled to deformation temperature at a rate of 30 °C/s and held for 5 s to eliminate temperature gradient. Finally, it was stretched at various strain rates until fracture followed by quenching using high pressure argon. Thermoforming tests of a structural part is carried out using a self-designed mold. First, put the blanking sheets into a furnace preheated to the austenitizing temperature; then, hold for 5 min to ensure that the cementite dissolves and diffuses uniformly; finally, transfer the austenitized blanking sheets to the thermoforming die for stamping. The actual temperature of the blanking sheets before stamping is measured by an infrared thermometer. To obtain as much martensite as possible, the holding pressure is maintained until the measured temperature reduces to 200 °C. Room temperature tensile tests of the thermoformed structural part were carried out using an Instron 3369 test machine with a stretch rate of 1 mm/s. Microstructure observation was carried out by Olympus-DSX500 optical microscopy (OM). OM samples were cut from the gauge with a direction parallel to tensile axis, then mechanically ground and polished before etching with 4% Nital. The initial microstructure, which consists of ~84% equiaxial ferrite and ~16% homogeneous distributed pearlite (volume fraction), is shown in Figure 1b.

## 3. Results and Discussion

### 3.1. Thermal Stress Response Behavior

True stress–true strain curves of the studied 22MnB5 boron steel at various strain rates and temperatures are shown in Figure 2. Visibly, the flow stress behavior is strongly affected by deformation temperature and applied strain rate. Except for that at the strain rate of $0.01s^{-1}$ and temperature of 500 °C, the flow stress increases gradually with the increase in applied strain, then tends to be saturated after reaching its peak, and finally decreases until fracture. The deformation in the early stage is predominated by strain hardening, leading to a rapid increase in flow stress. With the increase in applied strain, DRV and DRX occur

successively, resulting in an increased dynamic softening. The dynamic softening affects the rate of increase in flow stress and the evolution behavior with applied strain lower than that of the peak stress, which then completely counteracts the strain hardening with further increase in applied strain. Therefore, the flow stress exhibits dynamic saturation after reaching the peak. In addition, the flow stress decreases with the increase in deformation temperature. Visibly, the effect of temperature on flow stress increases with the reduction of deformation temperature. As the deformation temperature decreases from 650 to 500 °C, the flow stress increases significantly. This is because the deformation temperature of 500 °C is in the bainite transformation range, and the bainite generated during thermal tension significantly increases the flow stress [21]. It is widely known that the increased deformation temperature leads to an increase in the kinetic energy obtained by atoms, which finally facilitates the DRV and DRX [22,23]. Furthermore, the strain required for dynamic softening decreases with increasing deformation temperature. Therefore, the strain required for dynamic equilibrium between strain hardening and dynamic softening reduces. In other word, the strain corresponding to the peak stress of the studied steel decreases with increasing the deformation temperature.

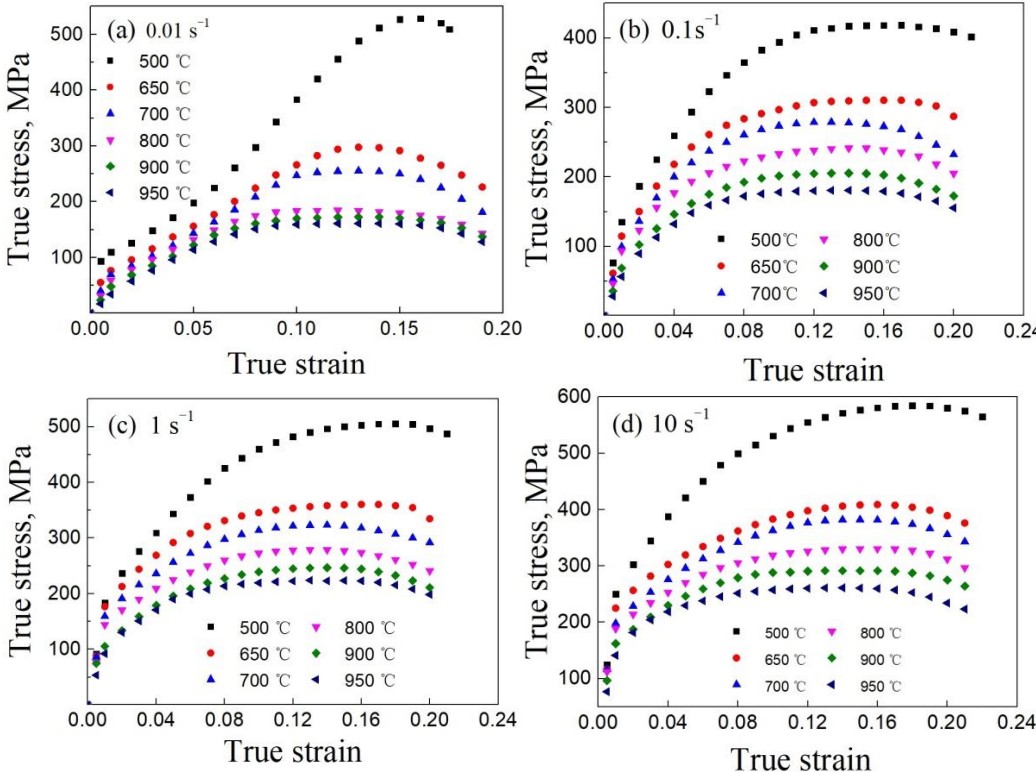

**Figure 2.** True stress true strain curves of the studied steel at strain rates of (**a**) 0.01 s$^{-1}$, (**b**) 0.1 s$^{-1}$, (**c**) 1 s$^{-1}$ and (**d**) 10 s$^{-1}$.

Except for that at the strain rate of 0.01 s$^{-1}$ and temperature of 500 °C, the flow stress of the studied 22MnB5 boron steel increases with the increase in applied strain rate, exhibiting apparent positive strain rate dependence, as shown in Figure 3. As reported, the increase in applied strain rate can effectively suppress dynamic softening and grain growth of metal materials by reducing response time for DRV and DRX [24,25], which can help to understand the positive strain rate dependence. In addition, an inflection point is detected when stretched at the strain rate of 0.01 s$^{-1}$ and temperature of 500 °C at which the slope of stress–strain curve exhibits a rapid increase. As a result, its peak stress reaches or even exceeds that at the strain rates of 0.1 and 1 s$^{-1}$. It is reported that, at low deformation temperature, the relatively low strain rate tension is beneficial to phase transition from supercooled austenite to bainite, and thus the steel is strengthened [11]. Therefore, the

flow stress of the studied 22MnB5 boron steel shows a sharp increase when stretched at the temperature of 500 °C and strain rate of 0.01 s$^{-1}$ (Figure 3a).

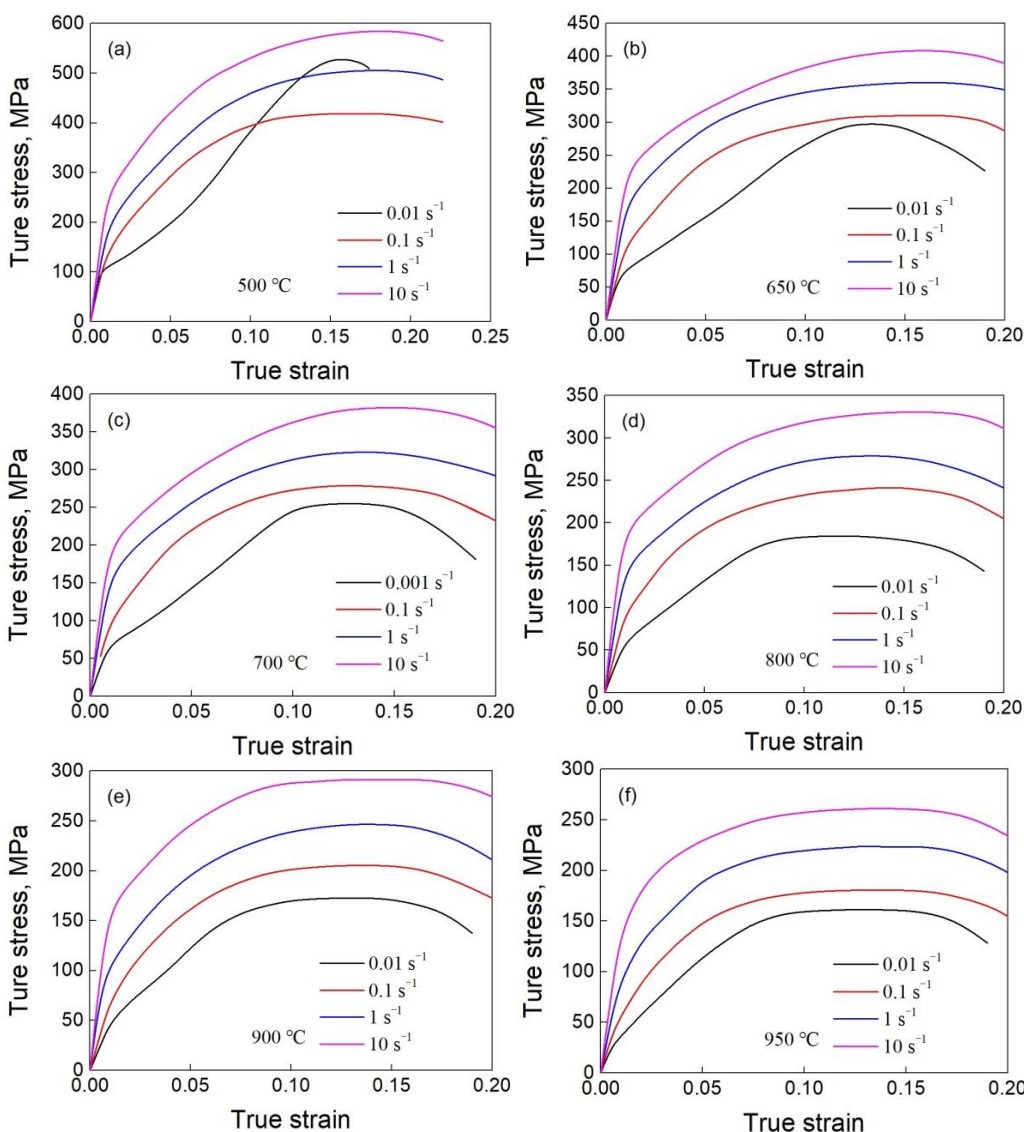

**Figure 3.** True stress–true strain curves of the studied steel at deformation temperatures of (**a**) 500 °C, (**b**) 650 °C, (**c**) 700 °C, (**d**) 800 °C, (**e**) 900 °C and (**f**) 950 °C.

The relationship between strain rate and peak stress at various deformation temperatures is shown in Figure 4. It can be observed that the peak stress decreases with increasing deformation temperature, which is attributed to dynamic softening resulted from the decreased interatomic force as well as the increased DRV and DRX. In addition, the peak stress increases with increasing applied strain rate when the deformation temperature is higher than 500 °C. It has been reported that, as the applied strain rate increases, the decreased response time weakens the DRV and DRX of materials [25], thereby resulting in an increase in peak stress. However, when stretched at 500 °C, the peak stress is characterized by decrease followed by continuous increase with increasing applied strain rate.

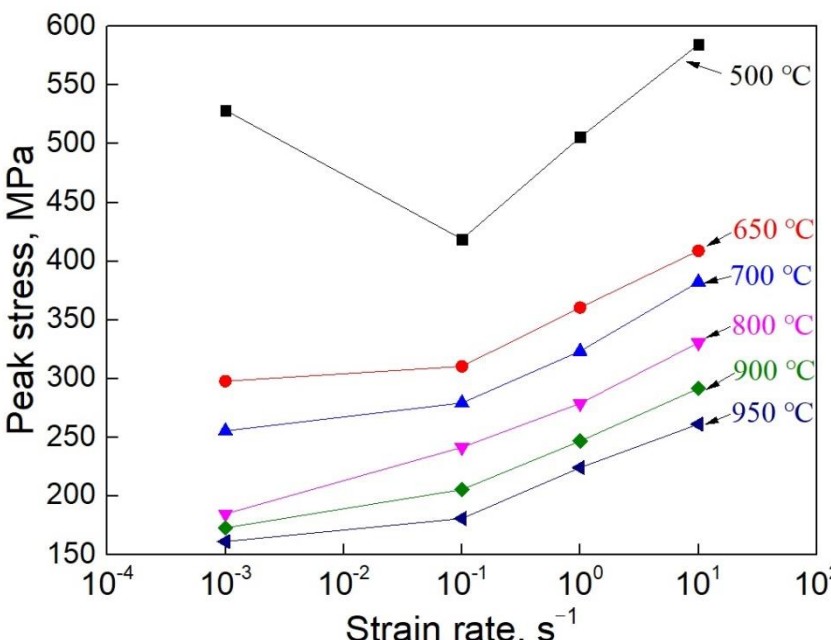

**Figure 4.** The relationship between strain rate and peak stress of the studied steel at various deformation temperatures.

The relationship between strain hardening rate (SHR) and true strain under various deformation conditions is obtained by differentiating the curves of true stress–true strain, as shown in Figure 5. At the same applied strain rate, the higher the deformation temperature, the lower the SHR. Additionally, the SHR at the same deformation temperature increases roughly with increasing applied strain rate. As reported, low applied strain rate facilitates phase transition from supercooled austenite to bainite as deformation temperature is in the bainite transition zone [26,27]. When stretched at the strain rate of 0.01 s$^{-1}$, the SHR shows a rapid decrease followed by an increase, and then it decreases again. The slope at the increasing stage drops apparently with the increase in deformation temperature. At the deformation temperatures of 500, 650 and 700 °C, a visible increase in SHR is detected, which is related to bainite transition because the deformation temperature is in the bainite transition zone. When the applied strain rate is higher than or equal to 0.1 s$^{-1}$, the SHR of the studied 22MnB5 boron steel shows a sharp decrease at initial deformation stage, and then shows a decrease with the increase in applied strain. The relatively low SHR in the middle and later deformation stages is related to DRV and DRX, leading to an increased dynamic softening with the increase in applied strain. Finally, a dynamic equilibrium is reached between dynamic softening and strain hardening. For 22MnB5 boron steel, 500 °C is in the bainite transformation zone, and the transformation amount of bainite generated during thermal tensile process is positively related with the deformation duration [21]. Even if the cooling rate is higher than the critical quenching rate of 22MnB5 boron steel, the low strain rate prolongs the deformation time, which is conducive to the transformation of supercooled austenite into bainite, thereby promoting a sudden increase in the deformation resistance of the studied steel [21]. In contrast, at high strain rate, the austenite does not have time to transform. Therefore, at the deformation temperature of 500 °C, the peak stress first decreases and then increases with the increase in strain rate due to the influence of bainite transformation.

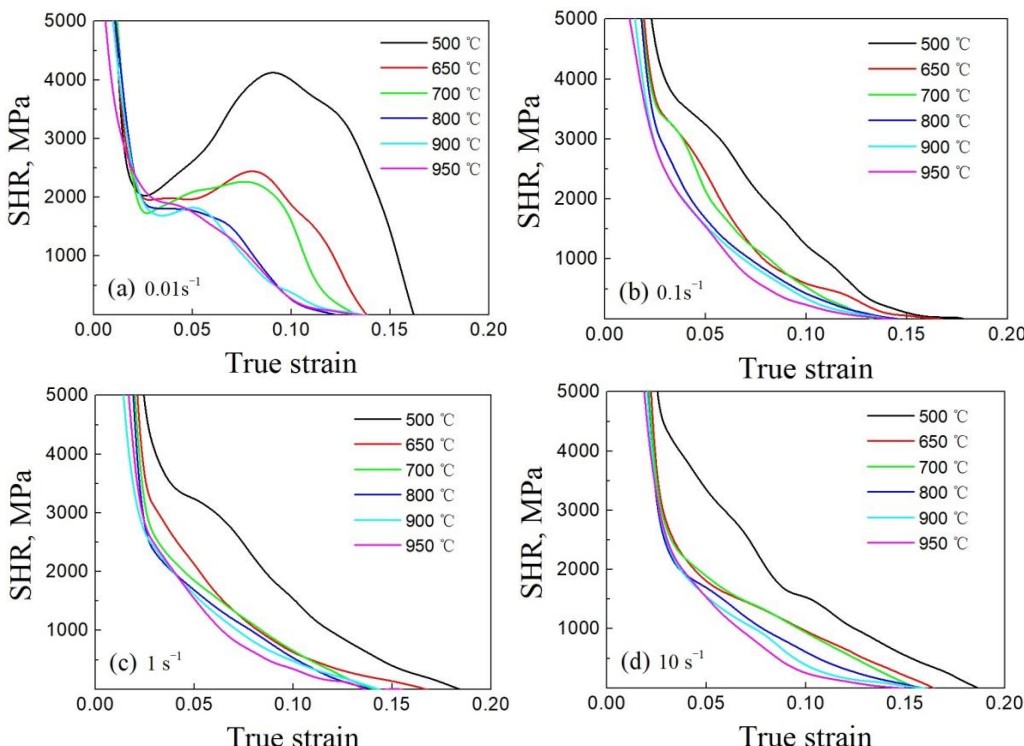

**Figure 5.** The relationship between SHR and true strain of the studied steel at stain rates of (**a**) 0.01 s$^{-1}$, (**b**) 0.1 s$^{-1}$, (**c**) 1 s$^{-1}$ and (**d**) 10 s$^{-1}$.

### 3.2. Microstructure Evolution

OM images with applied strain rate of 0.01 s$^{-1}$ at various deformation temperatures are shown in Figure 6, obtained from the gauge center (position 1) and the area near fracture surface (position 2). For 22MnB5 boron steel, the transformation temperature of eutectoid reaction (A$_{c1}$) is ~740 °C, and the phase transformation onset temperature from austenite to ferrite (A$_{c3}$) is ~860 °C [28,29]. In the present work, the samples were austenitized at 950 °C for 180 s before thermal tensile tests, and the following cooling rate 30 °C/s was higher than the critical quenching rate; therefore, bainite transition occurred when stretched at 500 °C. As shown in Figure 6a,b, a large number of bainites accompanied with ferrites and some pearlites are detected. As can be observed, relatively more bainites are observed at position 2 as compared with that at position 1. This is mainly due to the local plastic deformation induced by necking in the later deformation stage, which promotes the bainite transition [30,31]. As deformation temperature increases to 700 °C, a mixed microstructure of ferrite and lath-shaped martensite is detected, in which the ferrite is mainly distributed along grain boundaries. Ferrite transition occurs during thermal tensile test because the deformation temperature (700 °C) is in the range of ferrite transition. In contrast, martensite transition occurs during cooling process after tensile fracture. This can help to understand the two-phase mixed microstructure at 700 °C. It is well known that the temperature rise induced by plastic deformation inhibits ferrite transition, while it promotes martensite transition [32];thereby, apparently more martensite is detected in the necking region (position 2). When stretched at 800 °C, the microstructure is similar to that at 700 °C. In comparison, the volume fraction of lath-shaped martensite at 800 °C is higher.

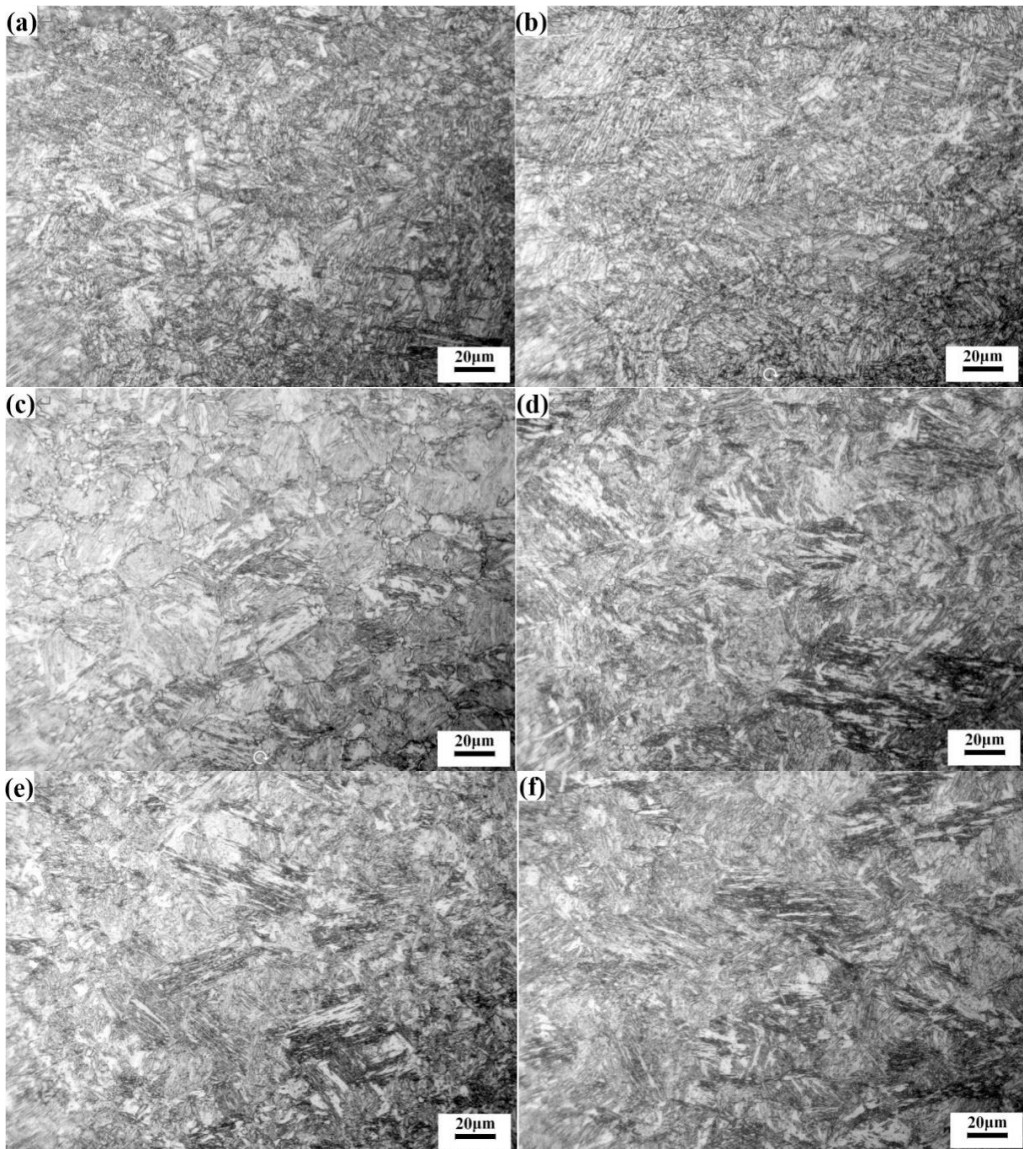

**Figure 6.** OM images of the fractured specimens at the specified positions: (**a**) position 1 with 500 °C; (**b**) position 2 with 500 °C; (**c**) position 1 with 700 °C; (**d**) position 2 with 700 °C; (**e**) position 1 with 800 °C; (**f**) position 2 with 800 °C.

Figure 7 shows OM images of the fractured samples at the position 1. The deformation temperature is 950 °C, and the strain rates for (a), (b), (c) and (d) are 0.01, 0.1, 1 and 10 s$^{-1}$ respectively. A typical lath-shaped martensitic is detected, indicating that the applied strain rate has little effect on martensite transformation. When stretched at 950 °C, the deformed microstructure at various strain rates is martensite accompanied by a small amount of residual austenite. In contrast, the size of the observed lath-shaped martensite decreases with the increase in applied strain rate, which is attributed to the fact that the relatively low applied strain rate is beneficial for the growth of austenite. At the strain rate of 10 s$^{-1}$, the austenitized grains are grow too late, such that the grain size in Figure 7d is basically the same as that of the undeformed grains in Figure 1b. Based on the above analysis, it can be concluded that the deformed microstructure of the studied 22MnB5 boron steel is closely related to thermoforming parameters, such as forming temperature and strain rate. A fine and lath-shaped martensite can be obtained by reasonable control of forming temperature and strain rate.

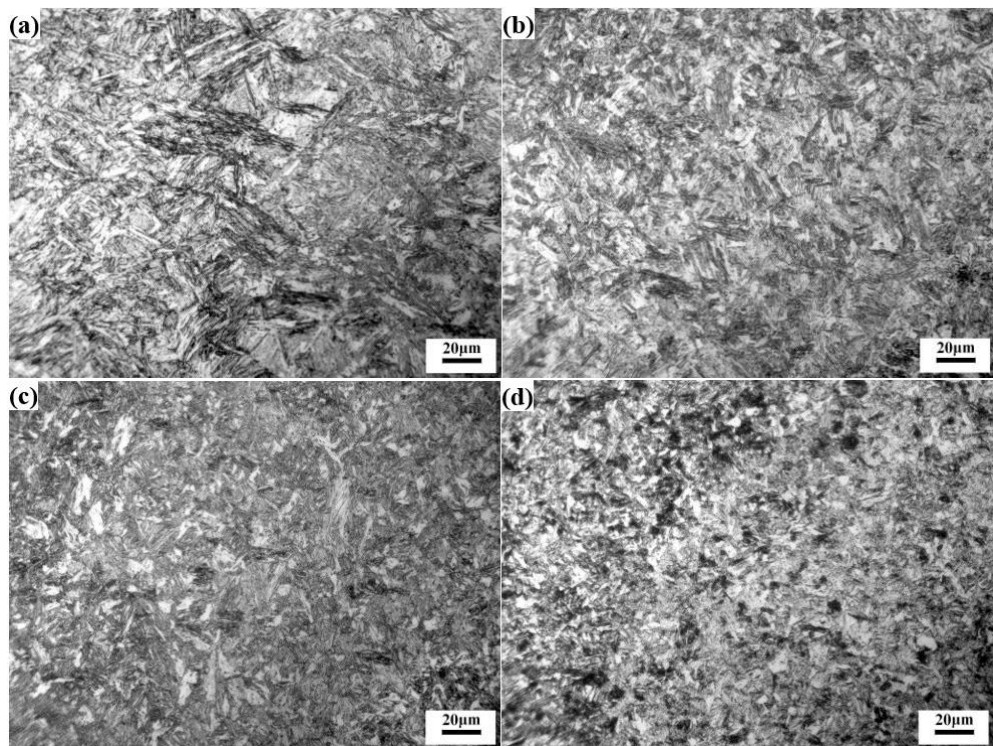

**Figure 7.** OM images of the fractured specimens at temperature of 950 °C with various strain rates: (**a**) 0.01 s$^{-1}$; (**b**) 0.1 s$^{-1}$; (**c**) 1 s$^{-1}$; (**d**) 10 s$^{-1}$.

### 3.3. Thermoforming of U-Shaped Structural Part

The studied 22MnB5 boron steel is widely used to manufacture the A-pillar, B-pillar and threshold beam of vehicle bodies. In the present work, the B-pillar is simplified to a U-shaped structure and taken as an example for thermoforming tests. There is no blank holder on the thermoforming die; therefore, two holes are designed on the blanking sheet for positioning. Dimensions (mm) of the blanking sheet and the U-shaped structure are shown in Figure 8. To achieve the cooling rate required for thermoforming, a water-cooling system is designed. The structure dimensions (mm) and cooling system layout of the thermoforming die are shown in Figure 9a,b. To obtain as much martensite as possible and ensure good formability for blanking sheet, the initial thermoforming temperature should be higher than 800 °C. In addition, heat loss is inevitable during the transfer from heat treatment furnace to thermoforming die. Therefore, 950 °C is selected as the austenitizing temperature. The thermoforming tests are performed with stamping velocity of 25 mm/s and holding pressure of 0.1 MPa. The average temperature of the blanking sheets before stamping is measured as ~851 °C. After holding pressure for 3 s, the average temperature drops to 390 °C, which is lower than the martensite-start temperature.

The thermoformed U-shaped structure is shown in Figure 10a. According to the characteristics of the U-shaped structure, four representative dimensions are selected to evaluate its forming accuracy, which are sheet thickness, total height, bottom width, and opening width. The measured results are listed in Table 2. Each value in Table 2 is the average of measurements from three U-shaped parts. For the sheet thickness, five points a~e on the U-shaped part are selected, as shown in Figure 8b. Generally, the dimensional accuracy of the thermoformed part is relatively high, and can meet the accuracy required for vehicle body structures. The sheet thickness is slightly reduced due to stretching effect during thermoforming. However, the thinning rate is very low and is almost negligible.

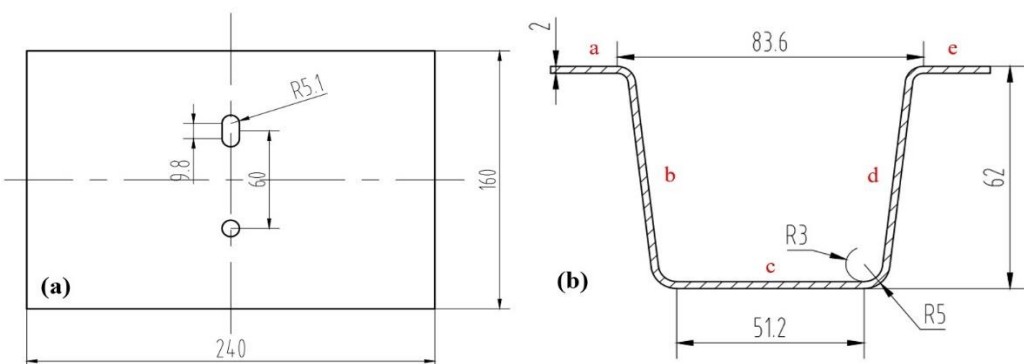

**Figure 8.** Dimensions (mm) of (**a**) blanking sheet and (**b**) U-shaped part.

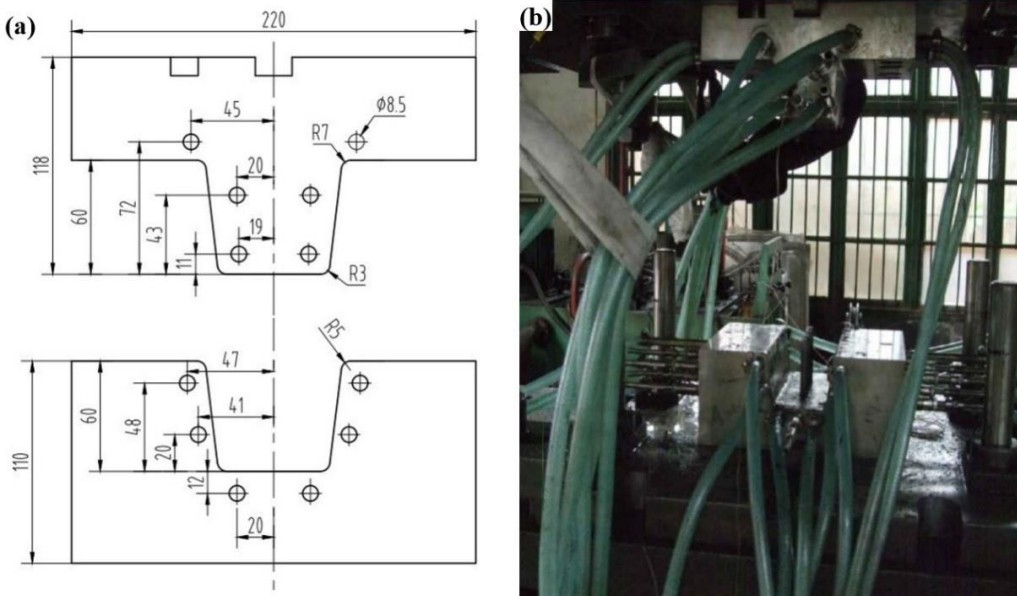

**Figure 9.** Schematics of (**a**) thermoforming die and (**b**) water-cooling system.

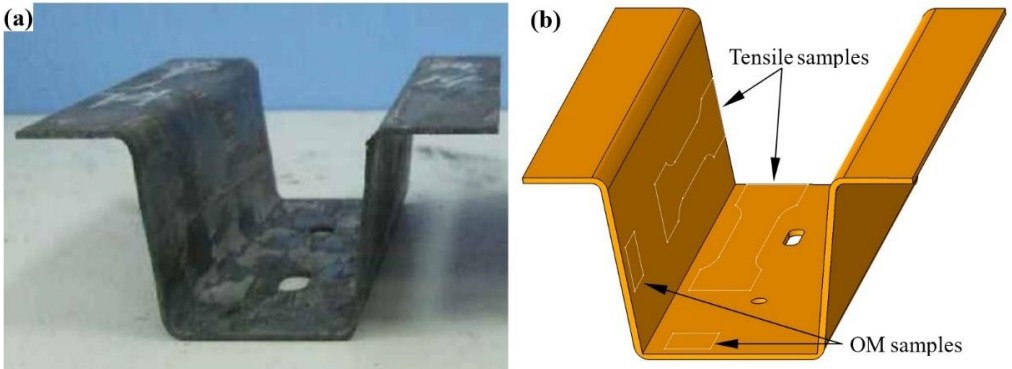

**Figure 10.** (**a**) Thermoformed U-shaped part and (**b**) schematic of sampling location.

**Table 2.** Averaged dimensions of the thermoformed U-shaped part.

| Sheet Thickness | | | | | Total Height | Bottom Width | Opening Width |
|---|---|---|---|---|---|---|---|
| a | b | c | d | e | | | |
| 20.1 | 1.94 | 1.99 | 1.95 | 2.01 | 61.97 | 50.52 | 84.61 |

To verify the performance of the U-shaped structure, samples for tensile test and microscopic observation were cut from the bottom and sidewall of the thermoformed U-shaped part. The sampling positions are shown in Figure 10b. The microstructures at the bottom and sidewall are shown in Figure 11. At the above two positions, a large amount of lath-shaped martensite is observed, accompanied by a small amount of residual austenite. In comparison, there is more residual austenite at the bottom, which indicates that the cooling rate at the sidewall is higher, and the quenching effect is better. The stress–strain curves of the two positions are shown in Figure 12, and their mechanical properties are listed in Table 3. The yield strength, ultimate strength and total elongation of the bottom position are 1036 MPa, 1245 MPa and 10.5%, and the mechanical properties corresponding to the sidewall are 1112 MPa, 1328 MPa and 9.7%. Compared with the as-received sheet, the yield strength and ultimate strength are significantly improved, while the total elongation is visibly reduced. As shown in Figure 1b, the initial microstructure of the as-received 22MnB5 boron steel is a mixture of ferrite and pearlite, so the yield strength and ultimate strength are relatively low, and the total elongation is good. However, the microstructure after thermoforming has been transformed into lath-shaped martensite (Figure 11), so the strength is greatly improved.

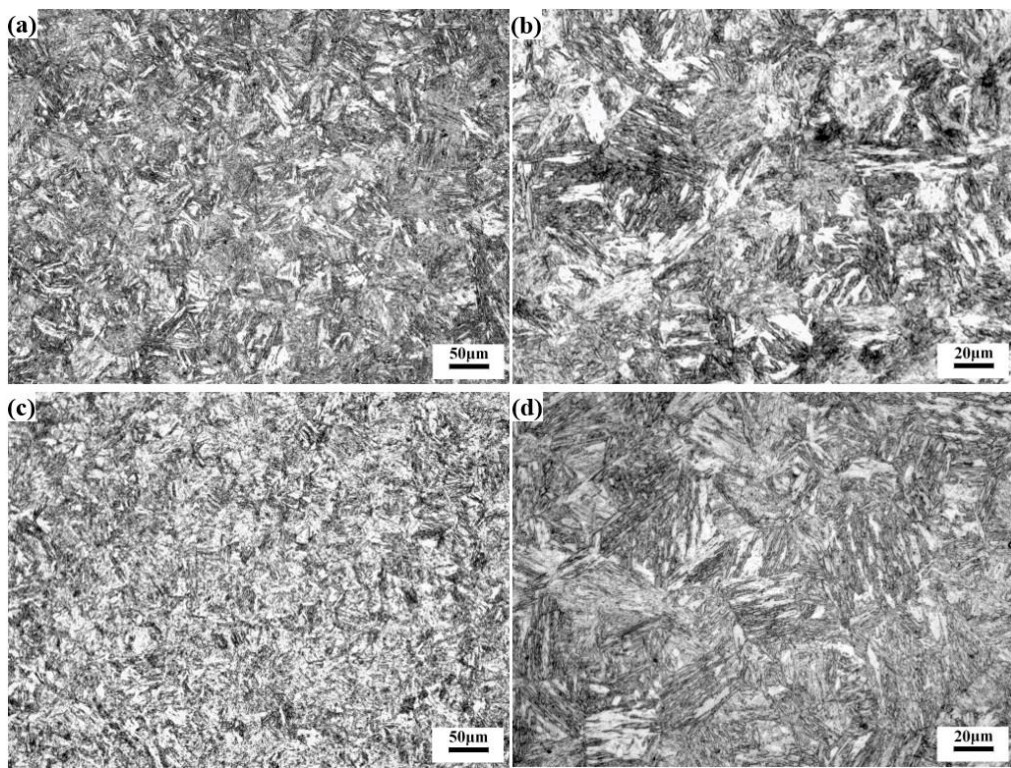

**Figure 11.** OM images at the bottom (**a**,**b**) and sidewall (**c**,**d**) of the U-shaped part.

**Table 3.** Tensile properties of the initial sheet and the thermoformed U-shaped part.

| State or Sampling Position | Yield Strength/MPa | Ultimate Strength/MPa | Total Elongation/% |
|---|---|---|---|
| Initial hot-rolled sheet | 416 | 571 | 30.0 |
| Bottom | 1036 | 1245 | 10.5 |
| Sidewall | 1112 | 1328 | 9.7 |

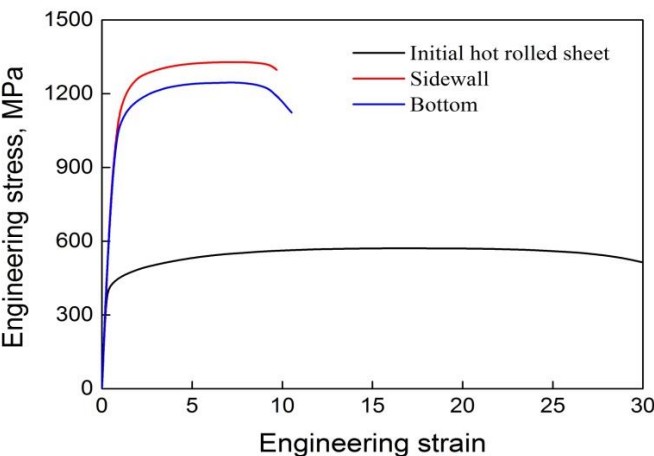

**Figure 12.** Stress–strain curves of the initial sheet and the thermoformed U-shaped part.

### 3.4. Constitutive Modeling

As shown in Figure 6, the bainite transition occurs when deformed at 500 °C with strain rate of 0.01 s$^{-1}$; therefore, only the stress–strain data at strain rates of 0.1, 1 and 10 s$^{-1}$ are selected for constitutive modeling. It is well known that the deformation at high temperature is a thermal activation process, and the dynamic equilibrium between strain hardening and thermal softening induced by DRV and DRX strongly affects the stress response behavior [33]. As shown in Figures 2 and 3, the stress response behavior of the studied 22MnB5 boron steel is apparently influenced by the applied strain rate and temperature. In present study, a modified Arrhenius model based on hyperbolic sine correction function is selected to describe the thermally activated deformation behavior of the studied 22MnB5 boron steel. This model contains the Zener-Hollomon parameter [34]; therefore, the flow stress is a function of deformation activation energy, deformation temperature and applied strain rate [35]. The expression is as follows:

$$\dot{\varepsilon} = Z \times \exp\left(-\frac{Q}{RT}\right) = A[sinh(B\sigma)]^n \exp\left(-\frac{Q}{RT}\right) \tag{1}$$

where $\dot{\varepsilon}$ is the strain rate (s$^{-1}$), $A$ is the structural factor, $B$ is the stress multiplier, $\sigma$ is the flow stress, $n$ is the stress exponent, $Q$ is the thermal deformation activation energy (kJ·mol$^{-1}$), $R$ is the gas constant (J·mol$^{-1}$·K$^{-1}$), $T$ is the deformation temperature (K) and $Z$ is the Zener-Hollomon parameter. Based on Equation (1), the following expression can be deduced:

$$Z = \dot{\varepsilon}\exp\left(\frac{Q}{RT}\right) = AF(\sigma) \tag{2}$$

where $F(\sigma)$ is the stress function, and can be expressed using the following formula.

$$F(\sigma) = [sinh(B\sigma)]^n \tag{3}$$

Equation (1) can be derived using Taylor series expansion. According to Taylor series expansion, Equation(1) can be simplified to a power function relationship when the flow stress is low ($B\sigma < 0.8$), as shown in Equation (4), and can be simplified to an exponential function relationship when the flow stress is high ($B\sigma < 1.2$) [36,37]. These expressions are shown in Equations (4) and Equation (5), respectively. Accordingly, Equation (4) is used to describe the creep process of metals, and Equation (5) can better depict the deformation of materials under high strain rate load.

$$F(\sigma) = \sigma^{m_1} \tag{4}$$

$$F(\sigma) = \exp(m_2\sigma) \tag{5}$$

where $m_1$ and $m_2$ are material parameters that satisfy the following relationship with $B$.

$$B = \frac{m_2}{m_1} \tag{6}$$

By combining Equations (1) and (4) and Equations (1) and (5), the following equations can be achieved.

$$ln\sigma = \frac{ln\dot{\varepsilon}}{m_1} - \frac{lnA}{m_1} + \frac{Q}{m_1 RT} \tag{7}$$

$$\sigma = \frac{ln\dot{\varepsilon}}{m_2} - \frac{lnA}{m_2} + \frac{Q}{m_2 RT} \tag{8}$$

The curves of $ln\sigma - ln\dot{\varepsilon}$ and $\sigma - ln\dot{\varepsilon}$ are plotted based on Equations (7) and (8). Thereby, the parameter $B$ can be determined by solving the slope of the curves. Treating the deformation activation energy $Q$ as a parameter that is unaffected by other conditions, the following equation can be built by combining Equations (4) and (5) followed by the natural logarithm.

$$ln\dot{\varepsilon} = lnA + nln[sinh(B\sigma)] - \frac{Q}{RT} \tag{9}$$

On this basis, the relationship curves of $ln[sinh(B\sigma)]$ as functions of $ln\dot{\varepsilon}$ and $\frac{1}{T}$ are plotted, and the stress exponent $n$ and the structural factor $A$ are calculated by fitting the linear regression equations.

Taking the experimental data at strain of 0.08 with temperature ranging from 500 °C to 950 °C as an example, the curves of $ln\sigma - ln\dot{\varepsilon}$ and $\sigma - ln\dot{\varepsilon}$ are drawn as shown in Figure 13. Clearly, each curve is basically linear with a similar slope. The slope and the correlation coefficient of regression analysis are listed in Table 4. By averaging the slope of each curve, the $1/m_1$ and $1/m_2$ are obtained with values of 0.0721 and 20.640. Finally, the parameter $B$ is calculated to be 0.0038 based on Equation (6).

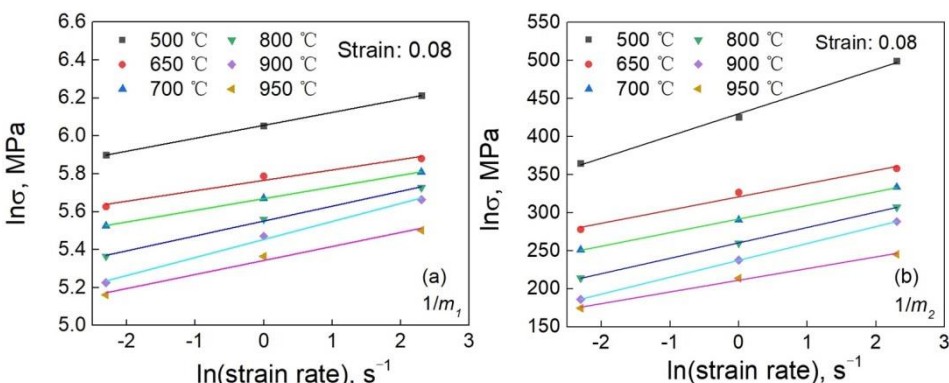

**Figure 13.** Curves of (**a**) $ln\sigma - ln\dot{\varepsilon}$ and (**b**) $\sigma - ln\dot{\varepsilon}$.

**Table 4.** Fitting results of $1/m_1$ and $1/m_2$.

| Deformation Temperature | $1/m_1$ | R | $1/m_2$ | R |
|---|---|---|---|---|
| 500 °C | 0.0682 | 0.9998 | 29.2132 | 0.9941 |
| 650 °C | 0.0550 | 0.9517 | 17.3797 | 0.9701 |
| 700 °C | 0.0615 | 0.9995 | 17.8688 | 0.9985 |
| 800 °C | 0.0785 | 0.9959 | 20.2343 | 0.9999 |
| 900 °C | 0.0951 | 0.9903 | 22.1823 | 0.9998 |
| 950 °C | 0.0740 | 0.9757 | 16.960 | 0.9920 |

The curves of $ln[sinh(B\sigma)] - ln\dot{\varepsilon}$ at various deformation temperatures are drawn and shown in Figure 14. Clearly, the $ln[sinh(B\sigma)]$ is approximately linear with $ln\dot{\varepsilon}$. The

correlation coefficients of linear fitting at 500, 650, 700, 800, 900 and 950 °C are 0.996, 0.981, 0.997, 0.996, 0.995 and 0.991, respectively. The value of $1/n$ at the strain of 0.08 is obtained by averaging the slope of each curve, so the parameter $n$ can be evaluated to be 9.666.

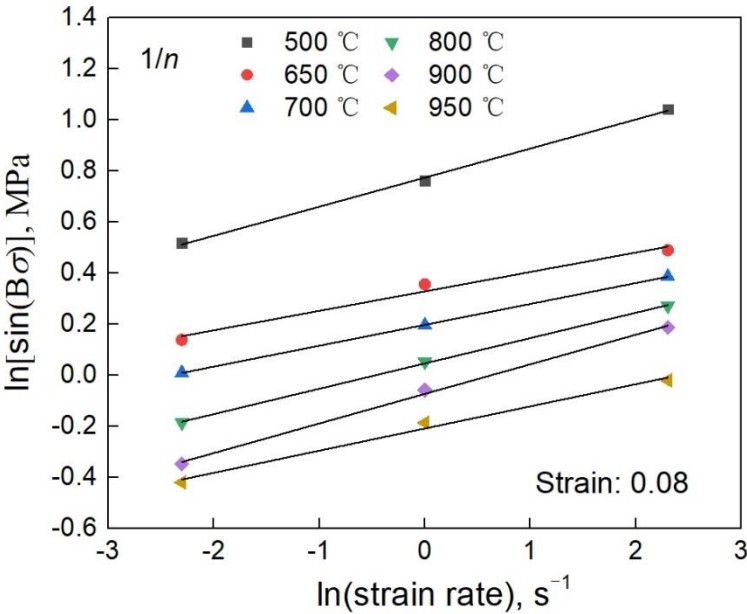

**Figure 14.** Relationship between $ln[sinh(B\sigma)]$ and $ln\dot\varepsilon$.

Equation (9) can be expressed as follows:

$$ln[sinh(B\sigma)] = \frac{Q}{nR} \times \frac{1}{T} + \frac{(ln\dot\varepsilon - lnA)}{n} \tag{10}$$

Based on Equation (10), the relationship between $ln[sinh(B\sigma)]$ and $\frac{1}{T}$ is drawn and shown in Figure 15. As can be seen, although a slight hyperbolic sine function relationship is shown, linear relationship is more consistent. Slope $k$, intercept $h$ and their correlation coefficient $R$ of linear fitting are obtained and listed in Table 5. Substituting them into Equations (11) and (12), the deformation activation energy $Q$ and $lnA$ are calculated as 196,778.5 kJ/mol and 19.894 s$^{-1}$, respectively.

$$Q = nRk \tag{11}$$

$$lnA = ln\dot\varepsilon - nh \tag{12}$$

**Table 5.** Linear fitting results of Figure 15.

| Strain Rate | Intercept $h$ | Slope $k$ | $R$ |
|---|---|---|---|
| 0.1 s$^{-1}$ | −1.479 | 1540.986 | 0.9473 |
| 1 s$^{-1}$ | −1.463 | 1367.931 | 0.9279 |
| 10 s$^{-1}$ | −1.513 | 1175.224 | 0.8708 |

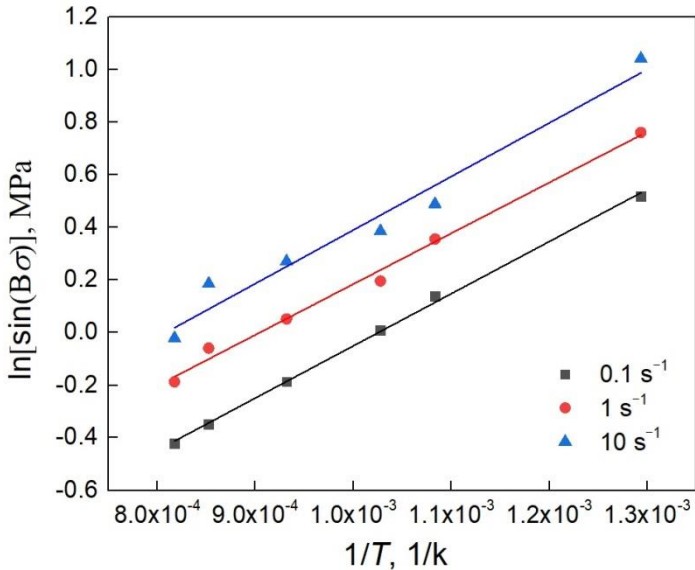

**Figure 15.** Relationship between $ln[sinh(B\sigma)]$ and $\frac{1}{T}$.

According to the above solving method, the $m_1$, $m_2$, $B$, $n$, $Q$ and $lnA$ at the strains of 0.02, 0.04, 0.06, 0.10 and 0.12 are calculated and listed in Table 6. Thereby, the regression equations are obtained by polynomial fitting, as shown in Equations (13)~(18). The fitting correlation coefficients are between 0.9954 and 0.9999, as shown in Table 7.

$$m_2 = 7.442 - 90.019\varepsilon + 5385.668\varepsilon^2 - 52907.628\varepsilon^3 + 155603.125\varepsilon^4 \tag{13}$$

$$m_1 = 0.052 - 0.817\varepsilon + 21.641\varepsilon^2 - 192.957\varepsilon^3 + 561.572\varepsilon^4 \tag{14}$$

$$B = 0.00789 - 0.132\varepsilon + 1.517\varepsilon^2 - 8.376\varepsilon^3 + 18.209\varepsilon^4 \tag{15}$$

$$n = 5.588 - 67.777\varepsilon + 4077.533\varepsilon^2 - 40203.068\varepsilon^3 + 118567.427\varepsilon^4 \tag{16}$$

$$Q = 90511 - 1729910\varepsilon + 73649500\varepsilon^2 - 618393000\varepsilon^3 + 1607990000\varepsilon^4 \tag{17}$$

$$lnA = 9.910 - 206.529\varepsilon + 8359.539\varepsilon^2 - 68579.966\varepsilon^3 + 174840.443\varepsilon^4 \tag{18}$$

**Table 6.** Fitting results of each parameter of the constitutive equation.

| Strain | $m_2$ | $m_1$ | $B$ | $n$ | $Q$ | $lnA$ |
|--------|-------|-------|-----|-----|-----|-------|
| 0.02 | 6.7467 | 0.0396 | 0.0058 | 5.073 | 118,590.5 | 12.022 |
| 0.04 | 8.6582 | 0.0399 | 0.0046 | 6.534 | 135,745.9 | 13.525 |
| 0.06 | 11.1464 | 0.0447 | 0.0041 | 8.434 | 168,524.3 | 16.953 |
| 0.08 | 13.7325 | 0.0528 | 0.0038 | 9.666 | 196,778.5 | 19.894 |
| 0.10 | 13.6682 | 0.0509 | 0.0037 | 10.351 | 210,094.6 | 21.196 |
| 0.12 | 14.2501 | 0.0529 | 0.0037 | 9.660 | 216,113.3 | 21.735 |

**Table 7.** Fitting correlation coefficient of the constitutive equation.

| | $m_2$ | $m_1$ | $B$ | $n$ | $Q$ | $lnA$ |
|---|-------|-------|-----|-----|-----|-------|
| $R$ | 0.9991 | 0.9983 | 0.9979 | 0.9954 | 0.9998 | 0.9999 |

Both sides of Equation (10) are taken logarithmically, and then simplified as follows:

$$\sigma = \frac{1}{B}\text{arcsinh}\left[\exp\left(\frac{ln\dot{\varepsilon} - lnA + Q/RT}{n}\right)\right] \tag{19}$$

Based on Equations (15)~(19), above, the flow stress constitutive equation of the studied 22MnB5 boron steel under isothermal deformation condition can be achieved. The comparison between constitutive fitted and experimental flow stresses is shown in Figure 16. Clearly, the constitutive fitting results are in good agreement with the experimental results, indicating that the constructed constitutive equation can accurately describe the thermal deformation behavior of the studied 22MnB5 boron steel. To more precisely characterize the fitting accuracy of the constitutive model, the experimental and fitting stresses at specific strains of 0.2, 0.4, 0.6 . . . 1.6 under different deformation conditions are selected for comparison. The results are shown in Figure 17. It is obvious that almost all the points are distributed within the error range of −5~5%, indicating that the established model has a high prediction performance in wide ranges of deformation temperature (500 °C to 950 °C) and strain rate (0.01 s$^{-1}$ to 10 s$^{-1}$).

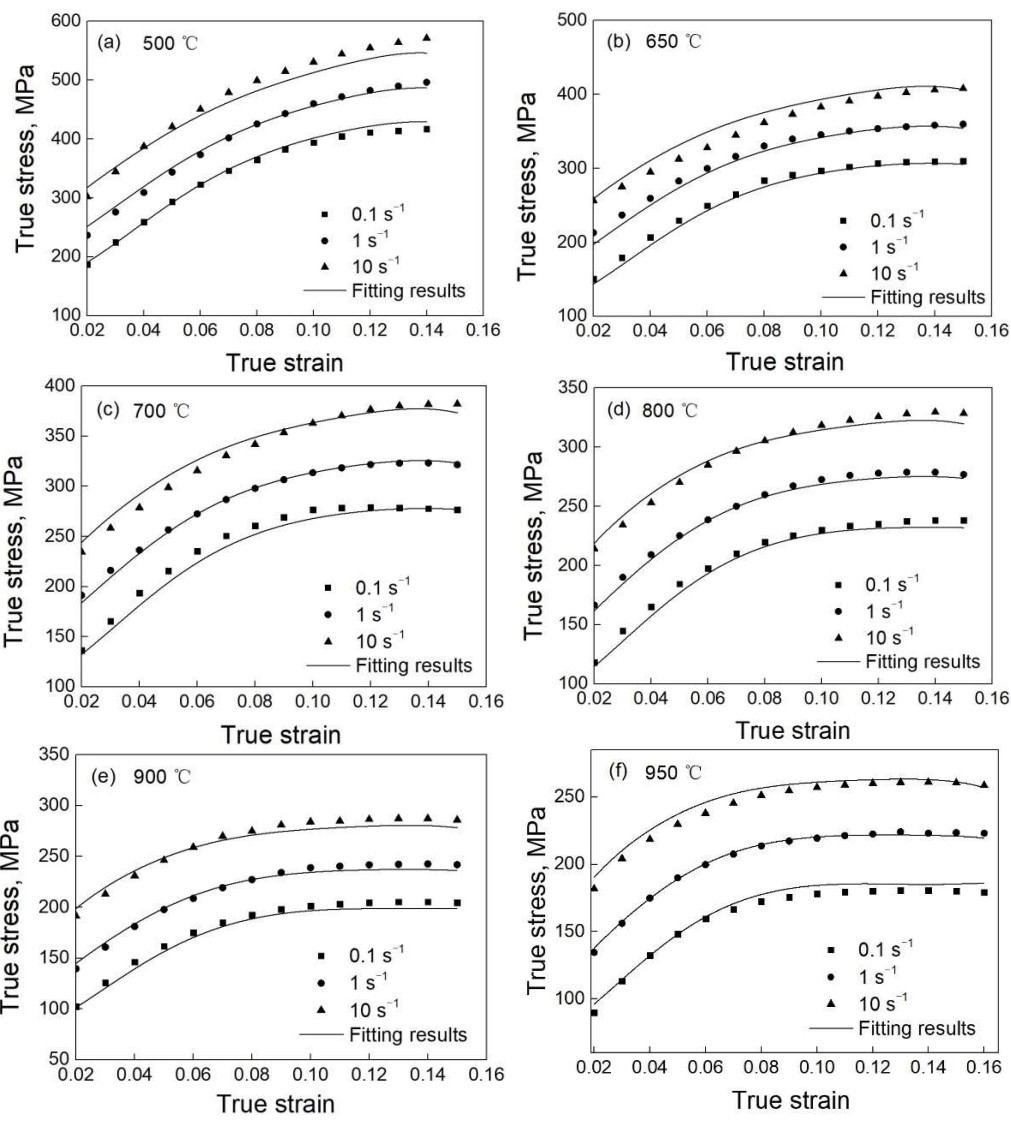

**Figure 16.** Comparison between constitutive fitting results and experimental results: (**a**) 500 °C; (**b**) 650 °C; (**c**) 700 °C; (**d**) 800 °C; (**e**) 900 °C and (**f**) 950 °C.

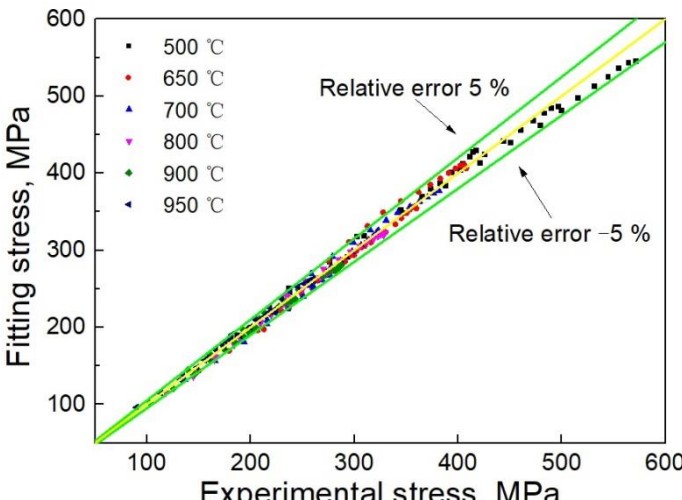

**Figure 17.** Correlation between experimental values and constitutive fitting values.

## 4. Conclusions

(1) Except for that at $0.01 \text{ s}^{-1}$ and 500 °C, the flow stress of the studied 22MnB5 boron steel increases gradually with the increase in applied strain and strain rate, exhibiting positive strain rate dependence. Then, it tends to saturation after reaching its peak. In contrast, it decreases monotonically with the increase in deformation temperature.

(2) With increasing deformation temperature, the microstructure changes from a mixture of bainite, ferrite and pearlite to lath-shaped martensite, ferrite and residual austenite, and finally to lath-shaped martensite accompanied by some residual austenite. The size of martensite decreases with the increase in applied strain rate.

(3) After thermoforming with an austenitizing temperature of 950 °C, the microstructures at the bottom and sidewall of the U-shaped part are lath-shaped martensite accompanied with some residual austenite, resulting in a significant increase in strength. In comparison, the strength of the sidewall is slightly higher than that of the bottom.

(4) Based on the Arrhenius constitutive model, a modified constitutive model that can precisely describe the thermal deformation behavior of the 22MnB5 boron steel is constructed with a relative error of less than 5%.

**Author Contributions:** Conceptualization, P.G.; methodology, P.G.; formal analysis, Q.Z. and P.G.; investigation, Q.Z., P.G. and F.Q.; data curation, P.G. and F.Q.; writing—original draft preparation, P.G.; writing—review and editing, P.G. and Q.Z. All authors have read and agreed to the published version of the manuscript.

**Funding:** This research was funded by the National Natural Science Foundation of China (grant number 51801051), the Scientific Research Project of Hunan Provincial Department of Education (grant number 18B193), the Youth Talent Support Program of Hebei Province (grant number BJ2020029) and the Handan Scientific Research Program (grant number 21422111276).

**Data Availability Statement:** The data presented in this study are available on request from the corresponding author.

**Conflicts of Interest:** The authors declare no conflict of interest.

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
