# Peer review of "Stress Response Behavior, Microstructure Evolution and Constitutive Modeling of 22MnB5 Boron Steel under Isothermal Tensile Load"

_metals, doi:10.3390/met12060930_

Round 1
Reviewer 1 Report
The article titled "Stress response behavior, microstructure evolution and constitutive modeling of 22MnB5 boron steel under isothermal tensile load" examines the effect of temperature and strain rate on the mechanical properties of 22MnB5 boron steel and describes a constitutive model for estimating the mechanical deformation of this material when subjected to different temperatures. The article is pleasant to read. However, I will make some comments and suggestions for improvement:
The English of the article should be improved and revised
The abstract needs to be restructured, in its current form it seems like an excerpt from the main text. It needs to include a contextualization of the study.
The introduction should be strengthened by some hypotheses, the main conclusions and the scientific contribution compared to other studies in the field. The research question must be stated in this section.
Please indicate the thickness of the specimen in Figure 1(a). How were the dimensions of the specimen determined? Were they determined based on a standard? Which standard?
Correct the position of a), b), etc. in the sub-figures in all figures according to the instructions in the template.
The article does not seem to be well organized. In section "2 Experiments" the experimental tests are described, then in section "3 Results and Discussion", more precisely in section 3.3 another experiment is described... The article needs to be restructured so that the information is not mixed. All experiments must be described in the same section.
Page 4, line 89 "Visibly, the flow stress behavior is determined by deformation temperature and applied strain rate" What is meant by "deformation temperature"? Please clarify this matter in the manuscript text.
In the caption of Figure 2, the dependence of the results on temperature must be stated.
Figure 5 shows the evolution of strain hardening rate (SHR) as a function of actual strain. Describe in the manuscript how the SHR was calculated/obtained.
On page 15, line 257, replace KJ with kJ.
On page 15, line 258, replace "serios" with "series".
In the conclusions, the authors state for the first time that they have adopted the Zenner-Hollomom factor, but nothing is said about this factor in the article. Describe in the manuscript how this factor was used in the proposed model
The authors did not discuss and interpret the results in light of previous studies, nor did they highlight the limitations of the study (the proposed model). Please add this information to the manuscript.
The conclusions should be updated with further findings, limitations, and future extensions.
Reviewer 2 Report
The manuscript “Stress response behavior, microstructure evolution and constitutive modeling of 22MnB5 boron steel under isothermal tensile load” carried out an analysis of the evolution of the microstructure and the flow stress behavior of a 22MnB5 steel as a function of temperature and strain rate.
For this purpose, authors carry out tests with different strain levels, strain rates and temperatures. Finally, a mathematical model has been proposed to explain the observed experimental behavior and it has been contrasted by comparing the experimental results with those obtained with the proposed model.
In my point of view, the paper is well structured. The introduction provides a wide compilation of the work made previously by others authors. The methods and experimental part is well explained and described, and the results are clear and very interesting. The conclusions resumes the work accomplished during the investigation.
I would like to recommend it for publication with the following minor considerations:
- The tests carried out for a strain rate of 0.01 s-1 and 500 ºC show a behavior, for most of the factors analyzed, different from that of the rest of the samples (lines 90, 107, 129…). It would be interesting to do a little deeper analysis that explains why this happens.
- Line 98: The authors state that "it is widely known." Some bibliographical reference should be included to endorse this statement.
- In several figures (fig 6, 7, 9) the signaling (a), (b)... is out of place. It must be placed correctly, on the image that it references.
- Line 176: “…in the position”. The position must be indicated.
- Figure 15: The units of the abscissa axis must be corrected
- Figure 16-c: It is not possible to distinguish the different series of the graph. Different symbols must be used for each series, as has been done in the rest of the graphs.

Reviewer 3 Report
Please find the attached file.

Reviewer 4 Report
This manuscript discusses the results of thermal tensile tests performed on hot rolled 22MnB5 boron steel. Their findings include the observation that the steel underwent a dramatic increase in tensile strength after thermoforming, which was due to the formation of a lath-shaped martensite accompanied by some residual austenite. Interesting article, although I have a few comments.
1. Please list what optical microscope was used to characterize the steel.
2. There are some issues with grammar in the manuscript that should be addressed. For example, "makes the kinetic energy obtained by atoms increase visibly" could be written in a more succinct manner.
3. In the Introduction section, the authors should discuss in more detail the deformation behavior observed in steels. Some articles below may be of use:
Mater. Sci. Eng. A-Struct. Mater. Prop. Microstruct. Process. 2022, 831, 8, Materials Science and Engineering: A 2019, 753, 135-145, Metallurgical and Materials Transactions A 2018, 49A, 147-161.
4. For Figure 2, please mention at which true strain value the inflection point occurs for the sample tested at 500 oC for a strain rate of 0.01 s-1.
5. In Figure 6, the lettering appears to be misaligned with the figure.
6. Please provide in an appendix the derivation for Eq. (2).
7. They also explain that Eq. (3) can be replaced by Eq. (4) under low stress conditions. Please define "low stress" in this context.
8. Please provide all the R values for Figure 14.
9. Also provide the goodness-of-fit values for Figure 16.
Round 2
Reviewer 1 Report
The authors have improved the article considerably. In my opinion, the article now meets the necessary requirements to be published in Metals. In view of this, I believe that the article should be accepted for publication.
Reviewer 3 Report
The issues raised by the Reviewer have been sufficiently addressed in the revised paper. However, a few minor issues still left:
1- Point 18 – Please note that the activation energy of deformation should be 196778.5 J.mol-1 or 196.7 kJ.mol-1. The authors wrote ‘196778.5 391 kJ/mol‘, which is wrong.
2- Figure 15 - The dimension of absolute temperature (T) is always written with the capital letter ‘K.’ Correct ir, please.
Reviewer 4 Report
The authors have adequately addressed my comments.